# Ontology Design Metapattern for RelationType Role Composition

Utkarshani Jaimini[1], Ruwan Wickramarachchi[1], Cory Henson[2] and Amit Sheth[1]

[1]*Artificial Intelligence Institute, University of South Carolina, Columbia, SC, USA*

[2]*Bosch Center for Artificial Intelligence, Pittsburgh, PA, USA*

### Abstract

RelationType is a metapattern that specifies a property in a knowledge graph that directly links the head of a triple with the type of the tail. This metapattern is useful for knowledge graph link prediction tasks, specifically when one wants to predict the type of a linked entity rather than the entity instance itself. The RelationType metapattern serves as a template for future extensions of an ontology with more fine-grained domain information.

### Keywords

ontology design pattern, link prediction, metapattern, relation type

## 1. Introduction

RelationType is a metapattern that specifies a role composition for a common path in a knowledge graph linking the head of a triple with the type of the tail. Given a triple *<head, relationName, tail>* that links two instances (*head, tail*) through an object property (*relationName*) and another triple that links the *tail* with its associated class (*tailClass*), *<tail, rdf:type, tailClass>*, then this implies a *relationNameType* triple that directly links the *head* with the class of the *tail* (*tailClass*): *<head, relationNameType, tailClass>*.

$$< head, relationName, tail > (relation\ link) \tag{1}$$

$$< tail, rdf:type, tailClass > (type\ link) \tag{2}$$

$$< head, relationNameType, tailClass > (relationNameType\ link) \tag{3}$$

The RelationType metapattern is useful for many practical knowledge graph (KG) link prediction applications. KG link prediction is the task of predicting new links in a KG using a machine learning model trained on the existing links. For example, given a head node (*head*) and an object property (*relationName*) in the KG, a link prediction model can predict a tail node (*tail*) such that the triple *<head, relationName, tail>* has a high likelihood of representing a true link in the KG. Link prediction algorithms are becoming more widespread and useful for various

*WOP'24: Workshop on Ontology Designs and Patterns, ISWC 2024,November 11-15, 2024. Maryland, USA*

✉ ujaimini@email.sc.edu (U. Jaimini); ruwan@email.sc.edu (R. Wickramarachchi); cory.henson@us.bosch.com (C. Henson); amit@sc.edu (A. Sheth)

ⓘ 0000-0002-1168-06843 (U. Jaimini); 0000-0001-5810-1849 (R. Wickramarachchi); 0000-0003-3875-3705 (C. Henson); 0000-0002-0021-5293 (A. Sheth)

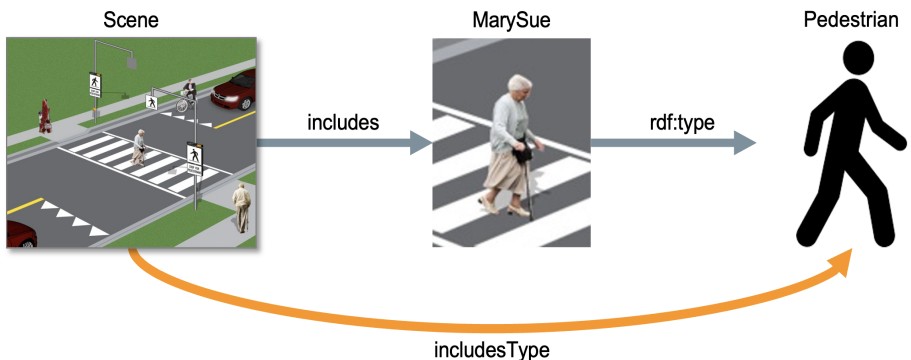

**Figure 1:** The image on the left represents a driving scene with included entities of various types, such as a car, stop line marking, pedestrian, bicycle, etc. The RelationType metapattern is used to directly link the scene instance node with the Pedestrian class node (i.e. with the includesType relation).

applications and tasks. In large KGs, with potentially millions of instance nodes, the search space for finding a correct node (e.g., *tail*) with link prediction is exceedingly large, resulting in a complex prediction problem. In practice, however, many practical use-cases require only predicting a type of a node rather than an instance node. For example, given the *scene* shown in Figure 1, we may be interested in predicting the type of entity (e.g. *Pedestrian*) included in the scene rather than predicting the specific entity instance (e.g. *MarySue*) included in the scene. Notice, however, that the *scene* node and the entity class nodes (e.g. *Pedestrian*) are not directly linked and therefore cannot be predicted by most KG link prediction algorithms. In this case, a solution to the issue would be to use the RelationType metapattern to create triples that directly link the *scene* instance node with the classes of entity nodes included in the scene, e.g.:



*<scene, includes, MarySue>* (relation link)
*<MarySue, rdf:type, Pedestrian>* (type link)
*<scene, includesType, Pedestrian>* (relationNameType link)



These additional *relationNameType* triples (e.g. *<scene, includesType, Pedestrian>*) can be used to train a KG link prediction model that can predict the entity types included in the scene. The proposed metapattern has been used across multiple use-cases for predicting entities in driving scenes, discussed in section 2.1 and for causal link prediction discussed in section 2.2

The paper is organized as follows: In section 2, a use case is introduced for a driving scene knowledge graph along with competency questions. The proposed RelationType metapattern is defined and discussed in section 3, including its primary concepts and axioms. Discussion of the RelationType metapattern along with future work and conclusion is outlined in section 4.

## 2. Use case: Driving Scene Knowledge Graph

We exemplified the RelationType metapattern using the driving scene knowledge graph (DSKG) [1]. An example of entity types present in the driving scenes include building, vegetation, car,

pedestrian, road signs, sidewalk, etc. Some of the entities have associated activity attributes, such as a car can be moving, parked, or stopped and a pedestrian can be walking, standing, or sitting, etc. The DSKG can be used for downstream task such as link prediction, entity prediction, scene clustering, scene classification, etc. In this paper, we will focus on the prediction task with competency question of form: *For a give entity (head of the triple) and its <relationName> predict the entity type of the tail of the triple.*

## 2.1. Predicting entities in a driving scene

Having to make decisions with incomplete information is an inherent problem in autonomous driving. While humans use their experiences and background knowledge to infer missing information, the autonomous car's ability to do so is only limited to its training data. For example, if not present in training data, computer vision models cannot yet understand that an observation of a ball in a residential scene may indicate the presence of a child following that ball. [1] formalized a solution for this as knowledge-based entity prediction (KEP) to predict the class of missing entities in DSKG using the RelationType metapattern. It allows answering the following types of competency questions in the entity prediction scenario:

CQ2.1.1  What unique *types* of entities are included in a given scene?

CQ2.1.2  What *additional* types of entities can be *predicted* that are not currently included in the scene (i.e., the KEP problem)?

Note that RelationType metapattern is especially useful when the prediction, inference, or querying task is performed directly on a sub-symbolic transformation of a KG (e.g., knowledge graph embedding). For example, when the above competency questions are modeled as KG link prediction tasks, the CQ2.1.1 corresponds to a *querying* task that seeks information about all unique entity types included in a scene, whereas CQ2.1.2 corresponds to a *prediction* task that provides *additional* information about potentially missing entity types.

## 2.2. Predicting causal entities in a scene

Humans have an intuitive understanding of the notion of causality in the world around us. In the autonomous driving scene, there exists a causal relation between a stop line marking, a pedestrian and a vehicle. A stop line marking causes a pedestrian to walk and cross the road, and causes a vehicle to come to a stop. The DSKG can be enriched with the above causal concepts and relations among the entities. In a downstream task of prediction, we are interested in predicting the type of the causal entity given a causal relation. The below competency questions take into account the existing causal relationship among the entities in Figure 1.

CQ 2.2.1  What is the *pedestrian activity* caused by a *stop line marking* in a scene?

CQ 2.2.2  What is the *pedestrian activity* caused by the *red car stopped* in a scene?

CQ 2.2.3  What is the *vehicle activity* caused by a *stop line marking* in a scene?

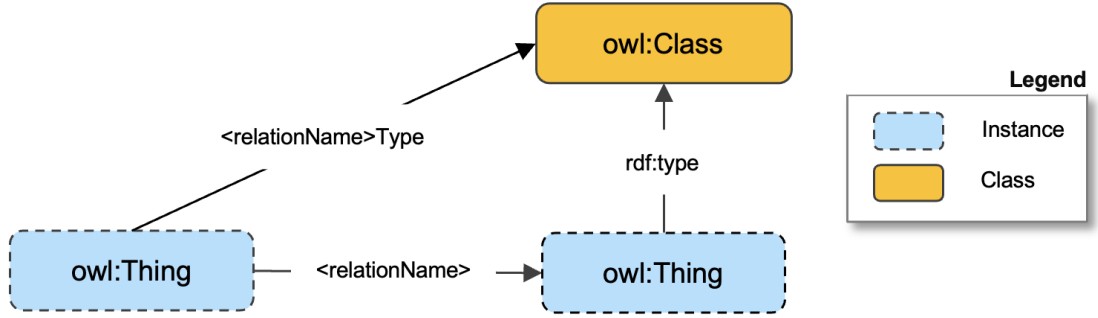

**Figure 2:** The RelationType metapattern illustrating the *<relationName>Type* link.

## 3. RelationType Metapattern

RelationType is a metapattern that represents a role composition, as shown in equations 4 and illustrated in Figure 2.

$$<relationName> \circ rdf:type \sqsubseteq <relationName>Type \qquad (4)$$

In this role composition, *relationName* is an object property linking the head and tail of a triple (i.e.*<head, relationName, tail>*), *rdf:type* is the RDF type[1] property linking the tail of the triple with its associated class (i.e.*<tail, rdf:type, tailClass>*), and *relationNameType* is a property linking the head of the triple with the class of the tail (i.e.*<tail, relationName-Type, tailClass>*). The scoped domain and range restrictions of these properties are defined below.

**relationName**: The head and tail of a *relationName* link can be an instance of any class.

$$\exists <relationName>.\top \sqsubseteq \top \qquad (ScopedDomain) \qquad (5)$$

$$\top \sqsubseteq \forall <relationName>.\top \qquad (ScopedRange) \qquad (6)$$

**relationNameType**: The head of a *relationNameType* link can be an instance of any class, the tail can be any class.

$$\exists <relationName>Type.owl:Class \sqsubseteq \top \qquad (ScopedDomain) \qquad (7)$$

$$\top \sqsubseteq \forall <relationName>Type.owl:Class \qquad (ScopedRange) \qquad (8)$$

Note that since RelationType is a metapattern, the defined axioms should be viewed as a template and not a concrete reusable component. In addition, *relationName* is a place holder for more concrete property names. To use this metapattern, in other words, one would need to instantiate it by replacing *relationName* in the axioms with a concrete property name. The *relationNameType* property is renamed by concatenating the concrete name of the *relationName*

---

[1]https://www.w3.org/TR/rdf12-schema/#ch_type

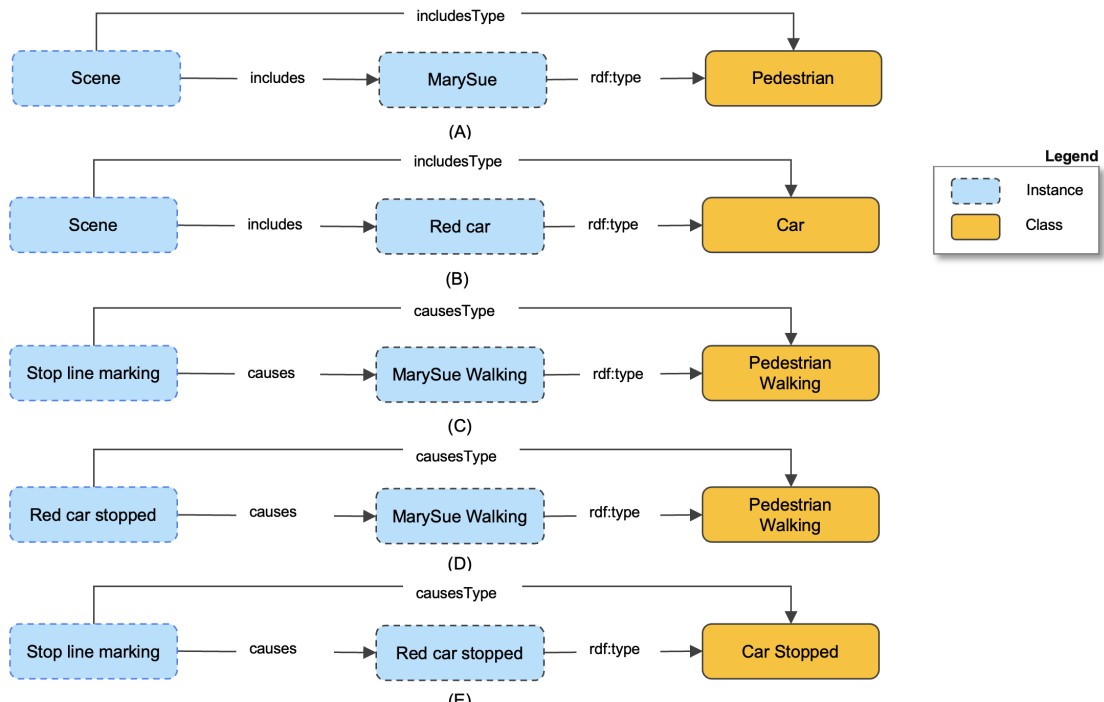

**Figure 3:** Illustrative diagrams of the RelationType metapattern used for the driving scene use cases, with the includesType and causesType relations.

with "*Type*". For example, in the use-case depicted in Figure 1 the *relationName* linking a scene with an entity is named "*includes*". So, in this case *relationName*Type is named as "*includesType*" (Figure 3).

The RelationType metapattern can be applied to links between instances of any type (i.e. of type `owl:Thing`[2]), including objects, places, agents, events, etc. The concept `owl:Thing` can be 1) an object that is composed of another object, 2) an event which includes sub events, 3) an agent which refers to a person or an organization. For the use case of a driving scene knowledge graph, the types of entities can include driving scenes, objects such as vehicles and stop line markings, agents such as pedestrians, and events such as walking pedestrians and stopping vehicles, etc. Figure 2 illustrates the RelationType metapattern. In this figure, two instances (of any class) are linked through the *relationName* property, and the tail of this link is associated with its class through *rdf:type*. The head of the link and the tail class are connected through the *relationNameType* property.

Figure 3 illustrates the RelationType metapattern for the driving scene use case (as shown in Figure 1). The *scene, red car stopped, stop line marking, MarySue walking* are instances of in the knowledge graph. In this case, *scene* is a context or situation in time and space with the presence of objects and occurence of events, *car* and *stop line marking* are objects, and *MarySue walking* is an event. The *Car, Pedestrian Walking* and *Car Stopped* are defined as classes (i.e. of

---

[2]https://www.w3.org/TR/2004/REC-owl-semantics-20040210/#owl _Thing

type `owl:Class`) and linked to their member instances using relation *rdf:type*, as shown in the Figure 3. An instance *scene* is connected to an instance *red car* using the *includes* property. The RelationType metapattern introduces the *includesType* property. This property can be used to train KG link prediction models in order to predict the type of entities included in a *scene*, such as a Pedestrian, Car, etc. and hence answering the competency questions in Section 2.1.

As another example, the *stop line marking* in a driving *scene* causes a *car to stop* momentarily. The *stop line marking* and a *stopped car* causes a pedestrian *MarySue* to walk across the street. Hence the instance of *stop line marking* is linked to the instance of *MarySue walking* and *red car stopped* using *<relationName>*, *causes*. The *relationNameType* property, *causesType*, is added to the above triples to help satisfy the competency questions in Section 2.2.

## 4. Discussion and Conclusion

A role composition ∘, used in RelationType metapattern, is a complex role inclusion axiom [2] which is a composition of two relations in description logic. It has been used to represent knowledge within complex structured domains such as medical terminology and chemical engineering [3, 4], which includes propagation relations such as part-whole, location, etc. At times, the complex domains require propagation of one property to another which can be introduced as path reification to be used for inference at a later stage.

The RelationType metapattern represents path using role composition axioms. RelationType is a type of metapattern which can have different instantiations for different class types depending on the use case. The metapattern was exemplified using a driving scene scenario. The RelationType is modular and generalizable metapattern. It can be further extended or reused by existing patterns which follows the above relational constraint.

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
