# OpenReview forum: "Ontology Design Metapattern for RelationType Role Compositions"
_swsa.semanticweb.org/ISWC/2024/Workshop/WOP — WOP 2024 Oral_

### Official Review · Reviewer_oGPA · 2024-08-21
**Review for Metapattern for RelationType role paper**

**Rating:** 7
**Confidence:** 3

**Review:**

Overview

This short paper the Metapattern called "RelationType" with one of its uses being stated as being in support of link/entity prediction tasks that is being increasingly deployed in Knowledge Graph based applications. In the paper, the concept of the RelationType metapattern is  described and a driving scene case study is presented to illustrate how the metapattern is used for both entity prediction and causal entity prediction.

Strengths

S1 The RelationType metapattern in the paper is well described, using clear language to illustrate the motivation for the metapattern, clear description of the metapattern itself,  and illustration of potential usage through a concrete intuitive case study
S2 The use of metapatterns and the general topic of their use, and the RelationType metapattern in particular,  in support of machine learning tasks, should lead to interesting discussion at the workshop

Weaknesses

W1 Personally I would (as a reader) have preferred to see Section 3 and Section 2 swapped, that is the theory introduced first, and then illustrated with the case study. Not strictly necessary for a camera ready but worth considering.
W2 It may be a function of space, but it would be useful for the reader if some of the assertions that are made are backed up with appropriate references. e.g. "In large KGs, with potentially millions of instance nodes, the search space for finding a correct node (e.g., tail) with link prediction is exceedingly large, resulting in a complex prediction problem." and " In practice, however, many practical use-cases require only predicting a type of a node rather than an instance node."
W3 The reader of the paper is likely to have wanted to hear the following point that is made in the conclusions section,  discussed in more detail and illustrated through example: "The RelationType is modular and generalizable metapattern. It can be further extended or reused by existing patterns which follows the above relational constraint."

Overall assessment
Interesting, well written and well illustrated short paper that should lead to interesting discussion at the workshop

---

### Official Review · Reviewer_dJ6t · 2024-08-26
**RelationType metapattern ignores existing OWL 2 semantics and consequences**

**Rating:** 2
**Confidence:** 4

**Review:**

This paper introduces the idea of a RelationType metapattern, where given a property $P$ used as follows:

```
?s ?P ?o.
?o rdf:type ?C.
```

has a corresponding property $P'$ that has the following as a consequence:

```
?s ?P' ?C.
```

I believe that, as part of the pattern, the IRI for $P'$ is simply a matter of appending $Type$ to the end of the $P$ IRI, however it's not clearly communicated in the paper.
Since I could not find a way to annotate a relationship between $P$ and $P'$, I can only assume that the pattern expects this kind of IRI manipulation.
The paper claims that the utility of this approach allows for improved link prediction performance, especially when it is more useful to predict the type of an entity involved in a statement than the identity of a specific entity.
The paper also doesn't provide any evaluation of these claims beyond demonstration of the pattern in KG snippets.

It however does provide a clear illustration of the pattern as designed, barring the issue of IRI manipulation.

Pros:
The paper is mostly well written and clear as to its intent and design. I feel confident that I could implement this pattern as intended in my own knowledge graphs.

Cons:
This paper has a few flaws that I think mar its quality and significance.

First, there is no discussion of related work that can help us understand the originality of the work. While it looks novel (for reasons I will discuss below), I cannot be sure of what the closest research is or what alternatives may exist to this approach.

This leads to the second issue, which is that there is a clear alternative to this approach that has been available in OWL through existential restrictions. To use the pattern above:

```
?s ?P ?o.
?o rdf:type ?C.
```

We can, instead of creating a new property $P'$, introduce an OWL property restriction as below:

```
?s rdf:type [ a owl:Restriction;
  owl:onProperty ?P;
  owl:someValuesFrom ?C
].
```

While it is slightly more verbose, it is compatible with existing standards and provides a form of reasoning support. If there is concern about the direct use of a restriction in this way, it can be expanded with an identified class:

```
?s rdf:type ?P'.

?P' a owl:Class;
  owl:equivalentClass [ a owl:Restriction;
    owl:onProperty ?P;
    owl:someValuesFrom ?C
  ].
```

However, these are semantically identical. While there may be limitations to this approach, since there has been no discussion of it in the paper, it is difficult to evaluate here. While I can see the issues around link prediction handling these sorts of constructs, it might be more productive to expand link prediction to support complex antecedents.

Another con is the A-box use of $C$ in the consequent. While I am an advocate of thoughtful uses of OWL 2 punning, I feel that this approach is not helpful or needed here.

Finally, I return to IRI manipulation. Again, I am forced to assume that appending $Type$ onto a given property as the approach, instead of some sort of annotation property linking them. This raises a number of issues around ontology design. URIs are intended to be completely semantically opaque, in fact many ontologies mandate the use of numeric codes in their URIs to make it easier to adjust metadata to fit referents, rather than needing to change URIs as language evolves or is translated. This metapattern approach violates that, which will preclude its use from most biomedical ontologies and many others. Further, it requires an out-of-band indication of semantics that a single annotation triple (`?P hasRelationType ?P'`) could remedy.

These issues significantly hinder the significance and quality of the work. Any changes to metamodel-level design requires significantly more consideration than is presented in this work.

---

### Official Review · Reviewer_5zhL · 2024-09-06
**Interesting metapattern for RDF graphs, but unclear relation to ontologies and ontology patterns**

**Rating:** 5
**Confidence:** 3

**Review:**

The paper introduces a so called metapattern, for representing “shortcuts” in an RDF graph, based on a property chain of an object property + a type relation. The authors state that this is quite useful in link prediction, since then the type can be predicted instead of specific instances, and exemplify it by the case of predicting types of things in a driving scene.

While the pattern is clear, and the motivating examples make sense, I have some doubts regarding the description of the pattern in the submitted paper. In particular in two respects; 1) the lack of connection to ontologies and ontology modelling, which is the main topic of the workshop, and 2) the lack of any empirical or even anecdotal evidence of the pattern usage, e.g. showing its claimed benefits compared to other alternatives.

Regarding point 1, the way the pattern is described in the paper makes it very clear that this is for KGs, not for ontologies. Since the pattern, if used with OWL, would lead to the need for punning, or some other work-around, since having a class as the object of an object property triple is otherwise not allowed. This may be warranted, and the pattern may still have its merits, but since this is not even mentioned, even less discussed, and analysed in terms of its consequences, it seems that this is more of a “data pattern” rather than an “ontology pattern”. Hence, the paper misses the one discussion that would be the most interesting for the WOP community, i.e. what does this imply in terms of OWL and ontology modelling?  Also, some alternative solution should have been discussed, e.g. other ontology design patterns that may provide alternative solutions. Overall, the paper totally lacks a discussion of related work, positioning the solution in relation to other existing work.

This leads to the second point, 2, which also includes the need to analyse and evaluate the usefulness of the pattern somehow. Although I recognise that it is very difficult, sometimes even impossible, to evaluate all aspects of an ontology design pattern, at least some anecdotal evidence should be provided. For instance, a concrete use case where this has been done, and where predictions were possible to make due to this pattern? Additionally, as already mentioned, at least some discussion of related work and alternative ways of modelling this should also have been discussed, to contrast this pattern against its alternatives.

Minor issues also include some repetition in the text, where more or less the same example is explained several times. While probably this is pedagogical in a sense, it is also a bit boring to the reader, since the pattern is not that complex. Instead the authors could have spent more text space on my points above, and on giving additional motivating examples/scenarios, i.e. showing a broader application of the pattern.

Overall, while this pattern would be interesting to discuss, adding the above points would extend the paper with probably several pages, and basically make it a whole new paper. Hence, I think it is probably too much additions required for acceptance.